# State Feedback and Deadbeat Predictive Repetitive Control of Three-Phase Z-Source Inverter

Fan Peng, Weicai Xie and Jiande Yan *

School of Electrical and Information Engineering, Hunan Institute of Engineering, Xiangtan 411104, China
* Correspondence: yjd@hnie.edu.cn

**Abstract:** In this paper, a composite control method combining repetitive control (RC) and deadbeat predictive control (DPC) is proposed to reduce the harmonic content of output voltage and improve the quality of voltage waveform, in order to solve the problem of voltage distortion caused by linear and nonlinear loads at the common grid-connected point of microgrid. First, the mathematical model of three-phase Z-source inverters is established, and the model is transformed into a state space expression. Then, Lyapunov's theory is used to find the design conditions of the state feedback control law based on linear matrix inequality. Finally, the parameters of the controller are solved by linear matrix inequality (LMI), and the parameter design of the improved repetitive controller is optimized. Furthermore, the system response speed is improved, and the system stability and robustness are guaranteed by combining the deadbeat predictive control technology. The simulation and experimental results verify the accuracy and superiority of the proposed deadbeat predictive repetitive control (DPRC) based on parameter optimization.

**Keywords:** Z-source inverter; microgrid Inverter; repetitive control; deadbeat predictive control

## 1. Introduction

In recent years, due to the shortage of traditional energy and environmental pollution, renewable energy has developed rapidly. At present, renewable energy is widely used in the microgrid composed of distributed generation systems. In micronets, as an important component of power and load connection, inverters play an important role in the stability of the system [1,2]. Energy conversion in microgrid is mainly carried out by power electronic devices. In order to promote the development of microgrid, it is necessary to ensure the efficient and stable operation of inverters. Most loads in the microgrid are non-linear. When the microgrid is operated as an isolated island, the output voltage of the inverter and the voltage of the public coupling point (PCC) will always be distorted due to the load harmonic current, resulting in serious stability and reliability problems [3,4]. The research of a control technique to lower output voltage total harmonic distortion (THD) in the circumstance of a non-linear load demand is therefore extremely important from a practical standpoint.

Z-source inverter (ZSI) is a new type of power converter. It can achieve boost function through the through state without dead-time [5]. Applying ZSI to a microgrid system can reduce the cost of inverters and output higher quality waveforms, and improve the reliability and energy conversion rate of the system [6,7]. Repetitive control can effectively track or suppress the periodic signal. By introducing a feedback link in the system, it enables the system parameters to track or suppress the periodic signal after continuous modification. Due to the lag link of single repetitive control and the limitation of its learning ability, many scholars usually combine repetitive control with other intelligent control methods to achieve the desired effect.

Traditional inverter control methods, such as proportional-integral (PI) control [8,9], are simple to design, but there will be steady-state error when the controlled amount is a

sinusoidal signal. Proportional-resonance (PR) controllers [10,11] can provide error-free control of AC signals, while repetitive control (RC) [12,13] can be seen as countless resonance controllers connected in parallel, thus enabling zero-static tracking of fundamental frequency signals and suppressing harmonic disturbances. However, when the repetitive controller is used alone, there is a delay of power frequency cycle and poor dynamic performance. Model predictive control [14,15] applied in power electronics can be divided into finite set model predictive control (FCS-MPC) and deadbeat predictive control (DPC). FCS-MPC chooses the best action according to the evaluation function and performance optimization index by establishing the system prediction model, but the switch state of single-phase inverter is very limited, which will cause large current fluctuation. deadbeat predictive control can calculate the optimal control quantity at the next moment according to the mathematical model of the system and is widely used in the control of grid-connected inverters. However, deadbeat predictive control depends on an accurate mathematical model, and control accuracy is affected by digital control delay and model parameter perturbation. Reference [16] on the basis of predictive control, the calculation delay compensation link and current reference correction link are added to effectively improve the quality of output power. Reference [17] by introducing sliding mode disturbance observer and induction parameter correction algorithm, inductance values are corrected in real time to enhance system robustness; Reference [18] an on-line algorithm to identify inductance parameters is added to the deadbeat predictive control, but it is necessary to inject pseudo-random binary sequence into duty cycle signal, which will affect the power quality of grid-connected power. Reference [19] proposed a method to smoothly predict output voltage while reducing current deviation constraints, which results in poor dynamic performance of the control system. Reference [20] proposes a composite controller which combines the advantages of RC and MPC and can effectively suppress periodic disturbances, but it requires a long adjustment time and has poor dynamic response capability.

Based on reference review, in order to solve the harmonic, reactive and PCC point power quality problems in three-phase, four-wire low voltage distribution network, this paper presents a composite control method based on parameter optimization, repetitive control and deadbeat prediction control, which is based on Z-source inverter and optimizes its controller as a whole. First, the microgrid circuit equation is transformed into a state space expression using a performance-preserving method. Secondly, state feedback control is introduced to transform the design of a repeating controller into an optimization problem with a set of LMI constraints. After that, deadbeat predictive control technology is implemented to give quick dynamic reaction following system startup or during significant load step changes, improving the system's control accuracy. The Lyapunov theory establishes the stability of the control strategy. Finally, a physical experiment platform is set up to verify the performance of the controller.

This paper is organized as follow: Section 2 introduces the Z-source inverter system and system modeling. Section 3 introduces the control strategy and stability analysis of the system. Section 4 will conduct simulation and experimental analysis. See Section 5 for the conclusion.

## 2. Three-Phase Z-Source Inverter Modeling

### 2.1. Fundamentals of Z-Source Inverters

Figure 1 below depicts the Z-source inverter's structure. Between the DC power supply and the inverter bridge, the Z-source network is introduced. The Z-source network includes mutually symmetrical impedance source networks (capacitors $C_1$ and $C_2$, inductors $L_1$ and $L_2$).

Z-source inverters can be opened at the same time on the switches above and below the same bridge arm. This situation is called through zero vector. Z-source inverters use the combination of through zero vector and traditional vector to achieve the purpose of pressure reduction and boost.

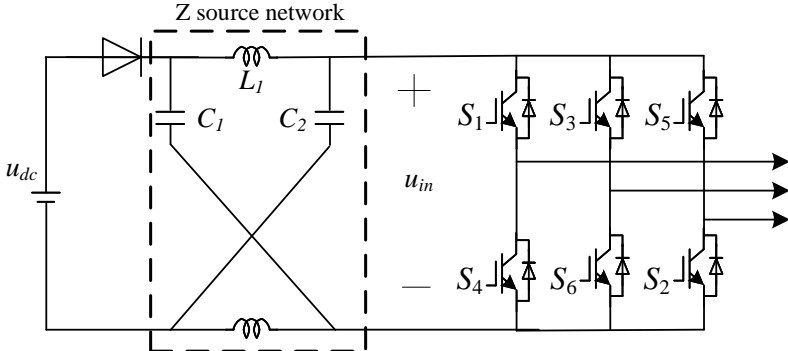

**Figure 1.** Z-source inverter topology.

Depending on whether the two power switches on the same arm are connected at the same time, the Z-source network can be classified as either straight state or non-direct state. Assume that the Z-source network is symmetrical for the sake of the discussion.

$$\begin{cases} L_1 = L_2 \\ C_1 = C_2 \end{cases} \tag{1}$$

In steady state, due to symmetrical circuit

$$\begin{cases} u_{L1} = u_{L2} = u_L \\ u_{C1} = u_{C2} = u_C \end{cases} \tag{2}$$

According to Figure 2, DC link voltage and input voltage is:

$$u_{dc} = u_d = u_{C2} + u_{L2} = u_{L1} + u_{C1} = u_L + u_c \tag{3}$$

$$u_{in} = u_{C2} - u_{Ll} = u_{Cl} + u_{L2} = u_C - u_L \tag{4}$$

According to Equations (3) and (4), the input voltage is as follow:

$$u_{in} = 2u_{C2} - u_{dc} \tag{5}$$

The Z-source inverter is in the direct state, which is the specific conduction state of the Z-source inverter, when two power switches on the same bridge arm and down are connected simultaneously. The DC power side diode is switched off, and the equivalent circuit diagram is shown in Figure 3.

From Figure 3, the following can be obtained:

$$u_{C1} = u_{C2} = u_C = u_{L1} = u_{L2} = u_L \tag{6}$$

At steady state, the bus voltage of the inverter is:

$$u_{in} = \frac{[(2u_C - u_{dc})T_1 + 0T_0]}{T_s} = u_C \tag{7}$$

where $T_s$ is a switch cycle of the switch, $T_0$ is the run time in the straight state, $T_1$ is the run time in the non-straight state.

The bus voltage $u_{in}$ of the inverter is in non-direct switching state

$$u_{in} = 2u_C - u_{dc} = \frac{T_s}{T_1 - T_0}u_{dc} = \frac{1}{1 - 2D_0}u_{dc} = Bu_{dc} \tag{8}$$

where $B$ is the pressor factor.

The peak $u_m$ and bus voltage of the output phase voltage of the three-phase inverter are satisfied

$$u_m = m\frac{u_{in}}{2} \tag{9}$$

Through the above equation, it can be concluded that the Z-source inverter can realize any value of output voltage without adding other conversion circuits, and can simultaneously reduce and boost the voltage. When the modulation degree is constant, the bus voltage decreases with the decrease of output voltage.

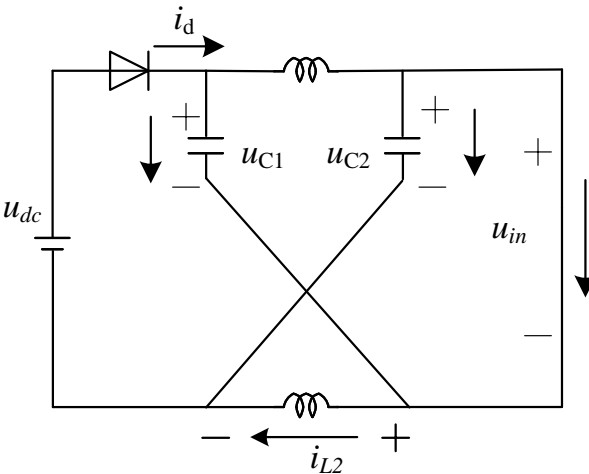

**Figure 2.** Equivalent circuit topology for non-direct Z-source Inverter.

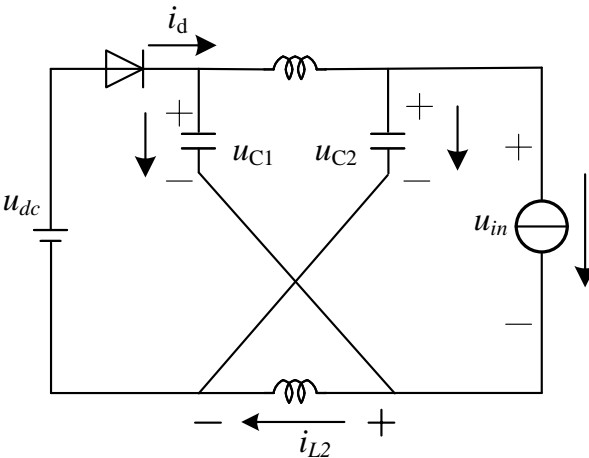

**Figure 3.** Equivalent circuit topology for direct Z-source Inverter.

### 2.2. Modeling of Three-Phase Z-Source Inverter

The topology of the three-phase microgrid Z-source inverter is shown in Figure 4 below. In the diagram, $u_{dc}$ is the input voltage, $u_{in}$ is the voltage after DC bus passes through Z-source network, $D_1$ is a diode, Z-source network includes capacitance $C_1$, $C_2$, inductance $L_1$, $L_2$. $L_f$ is the output filter inductance, $R_{Lf}$ is the equivalent internal resistance of inductive $L_f$, $C_f$ is the output filter capacitance, $u$ is the PWM control input voltage, PCC is the public connection point.

By decoupling capacitance and inductance, this study establishes the three-phase Z-source inverter state space equation. The filtered inductance current $i_L$ and filtered capacitance voltage $u_C$ of the decoupled d-axis are chosen as state variables, and the output of the controller is chosen as the input of the inverter. Since the parameters of the three-phase filter circuit are identical and the dq axes are independent of each other, the single-phase LC filter shown in Figure 5 is taken as an example for analysis. The effects of linear and non-linear loads on the controlled output voltage are modeled by the

indeterminate (time-varying) load admittance $Y_0(t)$ and the external current source $i_d(t)$. Figure 6 is the control structure of this paper.

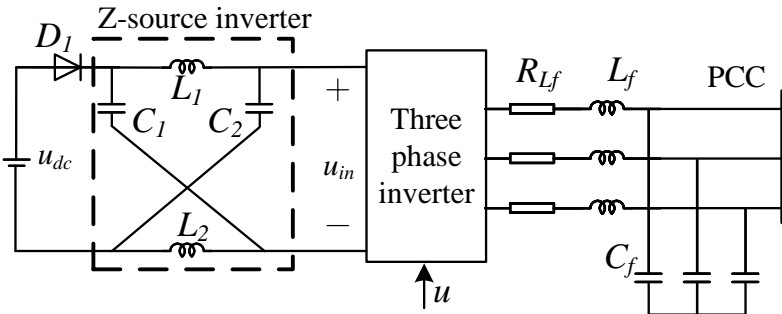

**Figure 4.** Topology of three-phase Z-source inverter.

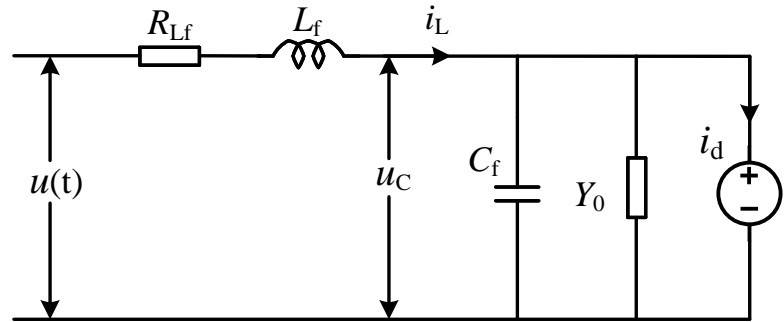

**Figure 5.** LC filter single-phase equivalent circuit.

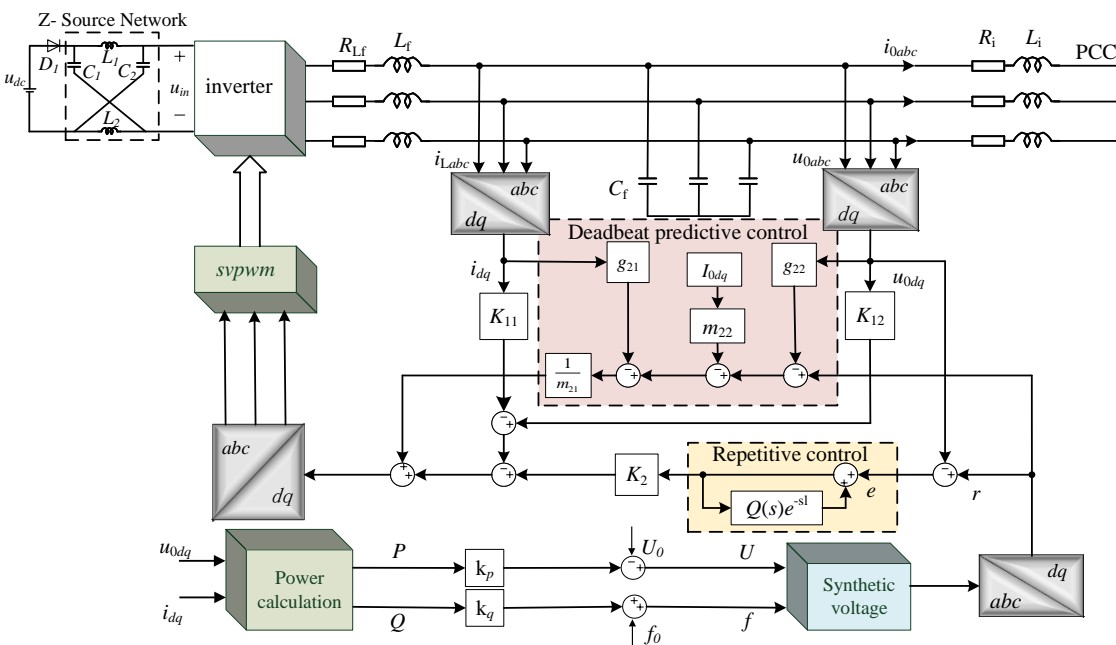

**Figure 6.** System control block diagram.

Based on Kirchhoff's Law, the circuit equation can be established as follow:

$$\begin{cases} u(t) = L_f \frac{di_L(t)}{dt} + R_{Lf} i_L(t) + u_C(t) \\ i_L(t) = C_f \frac{du_C(t)}{dt} + u_C(t) Y_0 + i_d(t) \end{cases} \tag{10}$$

Convert to a state space expression as follow:

$$\begin{bmatrix} \frac{di_\text{L}(t)}{dt} \\ \frac{du_\text{C}(t)}{dt} \end{bmatrix} = \begin{bmatrix} -\frac{R_\text{Lf}}{L_\text{f}} & -\frac{1}{L_\text{f}} \\ \frac{1}{C_\text{f}} & \frac{Y_0}{C_\text{f}} \end{bmatrix} \begin{bmatrix} i_L(t) \\ u_C(t) \end{bmatrix} + \begin{bmatrix} \frac{1}{L_\text{f}} \\ 0 \end{bmatrix} u(t) + \begin{bmatrix} 0 \\ \frac{1}{C_\text{f}} \end{bmatrix} i_\text{d}(t) \tag{11}$$

$$u(t) = \begin{bmatrix} 0 & 1 \end{bmatrix} \begin{bmatrix} i_L(t) \\ u_C(t) \end{bmatrix} \tag{12}$$

where $x(t)$ is the state vector of three-phase Z-source inverter, $x(t) = [i_\text{L}(t) \ u_\text{C}(t)]^\text{T}$, $i_L(t)$ is the inductance current, capacitance voltage, $y(t)$ is the control output, $u(t)$ is the control input, $i_d(t)$ is the periodic interference, $A(Y_0(t))$ is a matrix function of uncertain parameter $Y_0(t)$.

Assume that $Y_0(t)$ has a maximum and minimum value and is known, we can obtain the following:

$$Y_\text{min} \le Y_0(t) \le Y_\text{max} \tag{13}$$

$$Y_0(t) = Y_\text{N} + \delta(t)Y_\text{D}, \delta(t) \in [-1,1] \tag{14}$$

where

$$Y_\text{N} = \frac{Y_\text{min} + Y_\text{max}}{2}, Y_\text{D} = \frac{Y_\text{min} - Y_\text{max}}{2}$$

Through the Linear Fractional Transformation (LFT) method, the above equation can be described as:

$$A(Y_0(t)) = A(Y_\text{N}) + H(Y_\text{D})\delta(t)E, \delta(t) \in [-1,1] \tag{15}$$

where $A(Y_N)$, $H(Y_D)$ and $E$ are constant matrices of uncertain structures.

## 3. Controller Design

### 3.1. Design of Repetitive Controller

Design Figure 7 shows an improved repetitive controller with a low-pass filter, setting the first-order low-pass filter $Q(s)$ to:

$$Q(s) = \frac{\omega_\text{c}}{s + \omega_\text{c}}, \omega_\text{c} = \frac{1}{T} \tag{16}$$

where $\omega_c$ and $T$ are the turning frequency and time constant of the first-order low-pass filter.

$$y_{rc}(s) = \frac{1}{1 - Q(s)e^{-s\tau}}e(s) \tag{17}$$

where $y_{rc}(s)$ and $e(s)$ are Laplace transforms of output $y_{rc}(t)$ and error $e(t)$.

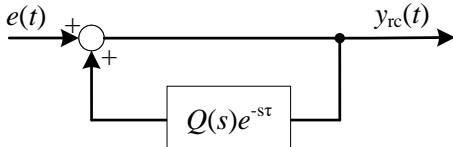

**Figure 7.** Improved repetitive control topology.

Convert Equation (17) to the state space expression as follow:

$$\begin{cases} \dot{x}_\text{rc}(t) = -\omega_\text{c}x_\text{rc}(t) + \omega_\text{c}x_\text{rc}(t-\tau) + \omega_\text{c}e(t-\tau) \\ y_{rc}(t) = x_{rc}(t) + e(t) \end{cases} \tag{18}$$

### 3.2. Design of State Feedback Controller

Considering the stability of the whole system, the following matrices are constructed:

$$Z(t) = \left[ \begin{array}{c} x(t) \\ x_{rc}(t) \end{array} \right] \in R^{(n+1)}$$

Equations (18) can be converted to:

$$\dot{Z}(t) = (A_a + \Delta A_a(t))Z(t) + A_d z(t - \tau) + B_a u(t) + B_q q(t) \tag{19}$$

where

$$q(t) = [r(t)i_d(t)]' \in R^2$$
$$A_a = \left[ \begin{array}{cc} A(Y_N) & 0_{n \times 1} \\ 0_{1 \times n} & -\omega_c \end{array} \right], \quad B_a = \left[ \begin{array}{c} B \\ 0 \end{array} \right]$$
$$A_d = \left[ \begin{array}{cc} 0_{n \times n} & 0_{n \times 1} \\ -C\omega_c & \omega_c \end{array} \right], \Delta A_a(t) = H_a \delta(t) E_a$$
$$H_a = \left[ \begin{array}{c} H(Y_D) \\ 0_{1 \times n} \end{array} \right], E_a = \left[ \begin{array}{c} E' \\ 0'_{1 \times n} \end{array} \right], B'_a = \left[ \begin{array}{cc} 0_{n \times 1} & B_d \\ \omega_c & 0 \end{array} \right]$$

According to Figure 8, the controller expression for a closed-loop system (19) can be converted as follow:

$$u_1(t) = K_1 x(t) + K_2 y_{rc}(t) \tag{20}$$

Further conversion to the state feedback expression is as follow:

$$u_1(t) = FZ(t) + K_2 r(t) \tag{21}$$

where $F \in R^{1 \times (n+1)} = [(K_1 - K_2 C)K_2]$.

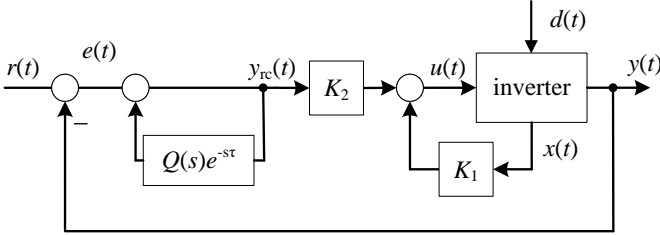

**Figure 8.** Repetitive control topology.

When combined with Equation (21), Equation (19) can be converted to:

$$\dot{Z}(t) = (A_\Delta + B_a F)Z(t) + A_0 Z(t - \tau) + \overline{B}_q q(t) \tag{22}$$

where $A_\Delta = A_a + \Delta A_a(t)$, $\overline{B}_q = \left[ \begin{array}{cc} BK_2 & B_d \\ \omega_c & 0 \end{array} \right]$.

To evaluate the above system's stability and make $q(t) = 0$, the closed loop system can be converted as follow:

$$\dot{Z}(t) = (A_\Delta + B_a F)Z(t) + A_d Z(t - \tau) \tag{23}$$

First, ensure that the system trajectory has the specified exponential decay rate $\alpha$

$$\|Z(t)\| \le \beta \|z(0)\| e^{-\alpha t}, t > 0 \tag{24}$$

Furthermore, the minimum cost function of the system is a measure of the transient performance of the system, as follow:

$$
\begin{cases}
J(p(t)) := \|p(t)\|_2^2 = \int_0^\infty p(t)'p(t)\mathrm{d}t \\
\qquad p(t) := C_p z(t) + D_p u(t)
\end{cases}
\tag{25}
$$

where $C_p$ and $D_p$ are constant matrices of appropriate dimensions.

Lemma 1 [21] Consider Equations (25) and (26), for a given positive scalar $\omega_c$ and $\alpha$, If there is a symmetric positive definite matrix $W, S \in R^{(n+1)\times(n+1)}$, matrix $Y \in R^{1\times(n+1)}$, regular scalar $\lambda$ and $v$ satisfy the following equation:

$$
\begin{bmatrix}
\Gamma(W,S,v) & e^{\alpha t}A_d W & WE_a & WC_\mathrm{p}' + Y'D_\mathrm{p}' \\
e^{\alpha t}WA_d' & -S & 0_{(n+1)\times(n+1)} & 0_{(n+1)\times 1} \\
E_a'W & 0_{(n+1)\times(n+1)} & -vI_{(n+1)} & 0_{(n+1)\times 1} \\
C_\mathrm{p}W + D_\mathrm{p}Y & 0_{1\times(n+1)} & 0_{1\times(n+1)} & -\lambda
\end{bmatrix} < 0
\tag{26}
$$

where $\Gamma(W,S,v) = A_a W + WA_a' + 2\alpha W + B_a Y + Y'B_a' \, S + vH_a H_a'$

If the above inequality is established, the closed-loop system is asymptotically stable when $F = YW^{-1}$. At the same time, the gain of the system can be obtained as follow:

$$
F = YW^{-1} = \begin{bmatrix} F_1 & F_2 \end{bmatrix}
\tag{27}
$$

Therefore, the controller can be converted as follow:

$$
u_1(t) = K_{11}I_L(t) + K_{12}u_C(t) + K_2 y_{rc}(t)
\tag{28}
$$

where $K_2 = F_2$, $\begin{bmatrix} K_{11} & K_{12} \end{bmatrix} = F_1 + K_2 C$.

In this paper, LMI is used to optimize the controller parameter design, and the parameters used for solving are shown in Table 1. According to Lemma 1, the optimal solution of linear matrix inequality is obtained through MATLAB toolbox, and the parameters of the controller are:

$$
K_1 = [-181.23 \; -153.78]
$$
$$
K_2 = 5.24 \times 10^3
$$

**Table 1.** Relevant parameters set by the system.

| Parameter | Value |
|---|---|
| Maximum admittance/S | 0.2 |
| Minimum admittance/S | 0.0001 |
| Damping resistance $\omega_c$ | 1000 |
| Filter inductance $L_f$/mH | 0.6 |
| Damping resistance $\alpha$ | 154 |
| Filter capacitor $C_f$/$\mu$F | 1500 |
| Damping resistance $R_L$/$\Omega$ | 0.01 |
| DC bus voltage/V | 400 |
| Switching frequency $f$/KHz | 21.6 |

*3.3. State Feedback Deadbeat Predictive Repetitive Control*

Based on the state space formulation of Equation (16) and the equivalent impulse principle, the deadbeat predictive controller is only utilized to offer quick dynamic reaction during system startup or load step changes. To make the calculation easier, use $Y_0(t)$ as the fixed value to obtain:

$$
\begin{cases}
x(k+1) = Gx(k) + M_1 u_2(k) + M_2 i_d(k) \\
y(k) = Cx(k)
\end{cases}
\tag{29}
$$

where

$$G = e^{A(Y_0(t))T_s} = \begin{bmatrix} g_{11} & g_{12} \\ g_{21} & g_{22} \end{bmatrix}, x(k) = \begin{bmatrix} i_L(k) \\ u_C(k) \end{bmatrix}$$

$$M_1 = A(Y_0(t))^{-1}\left(e^{A(Y_0(t))T_s} - I\right)B = \begin{bmatrix} m_{11} & m_{12} \end{bmatrix}^T \tag{30}$$

$$M_2 = A(Y_0(t))^{-1}\left(e^{A(Y_0(t))T_s} - I\right)B_d = \begin{bmatrix} m_{21} & m_{22} \end{bmatrix}^T$$

Expand Equation (29) as follow:

$$\begin{cases} u_C(k+1) = g_{11}u_C(k) + g_{12}i_L(k) + m_{11}u_2(k) + m_{21}i_d(k) \\ i_L(k+1) = g_{21}u_C(k) + g_{22}i_L(k) + m_{12}u_2(k) + m_{22}i_d(k) \end{cases} \tag{31}$$

Assuming that $u_C(k)$ and $i_L(k)$ are known at $t = k$, the output voltage $u_2(k)$ at $t = k$ can be calculated according to Equation (31) as follow:

$$u_2(k) = \frac{1}{m_{11}}[u_C(k+1) - g_{11}u_C(k) - g_{12}i_L(k) - m_{21}i_L(k)] \tag{32}$$

In fact, from the above equation, as can be observed, the optimal reference voltage for a conventional sinusoidal inverter is its output voltage. Therefore, instead of $uc(k + 1)$ in the above formula, use reference directive $r(k + 1)$ as follow:

$$\begin{aligned} u_2(k) &= \frac{1}{m_{11}}[r(k+1) - g_{11}u_C(k) - g_{12}i_L(k) - m_{21}i_L(k)] \\ &= K_3 r(k+1) - K_4 u_C(k) - K_5 i_L(k) - K_6 i_0(k) \end{aligned} \tag{33}$$

Combined with Equation (28), the control law of the system can be obtained by repetitive the deadbeat predictive control structure as shown in Figure 9.

$$\begin{aligned} u_k(k) &= (K_{11} - K_5)i_L(k) + (K_{12} - K_4)u_C(k) \\ &\quad + K_2 y_{rc}(k) + K_3 r(k+1) - K_6 i_0(k) \end{aligned} \tag{34}$$

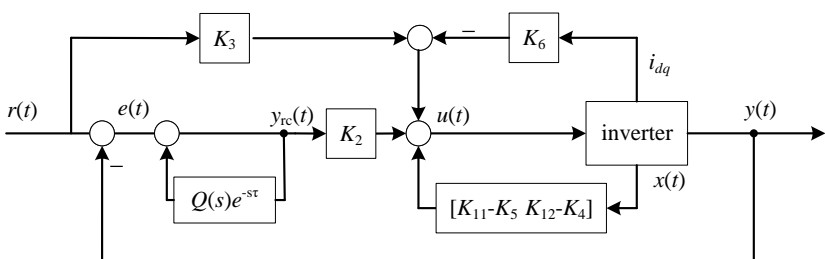

**Figure 9.** State feedback deadbeat predictive repetitive control structure.

## 4. Simulation and Experimental Verification

### 4.1. Simulation Results

On the MATLAB/Simulink software platform, this research creates a simulation model for microgrid functioning, as shown in Figure 10, which includes two DGs.

The optimum power supply maintains the DG module's DC bus voltage by utilizing the same LC filter and line impedance. The public load load3 is connected to the public AC bus of the microgrid, which is switched on and off by the switch, and the standard feeder impedance of the low-voltage microgrid is $0.584 + j0.0102\ \Omega/\text{km}$, taking the line impedance of DG1 and DG2 as $1.298 + j0.0174\ \Omega/\text{km}$. In the diagram, $u_{pcc}$ is used to define the PCC point voltage. This paper verifies the effectiveness of the proposed control strategy through simulation experiments. The simulation parameters are shown in Table 2.

**Table 2.** Relevant parameters set by the system.

| Parameter | Value |
|---|---|
| Rated frequency/Hz | 50 |
| Filter parameters | $w_{f1} = 50; w_{f2} = 100$ |
| Line impedance/$\Omega$ | 1.276+j 0.0146 |
| Voltage amplitude/V | 110 |
| Droop coefficient | $k_p = 10^{-5}, k_q = 3 \times 10^{-4}$ |

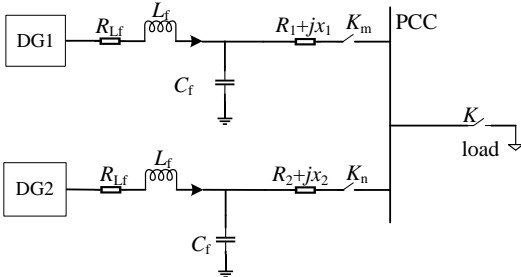

**Figure 10.** Microgrid simulation model.

The Z-source controller can be used to increase the input voltage. In order to better demonstrate the performance of the Z-source controller, the DC chain voltage was adjusted to be the same as a reference value of 175 V. Figure 11 shows the steady-state waveform of the Z-source parameter. The mean value of the inductive current is 1.72 A, and the reasonable ripple is 0.15 A.

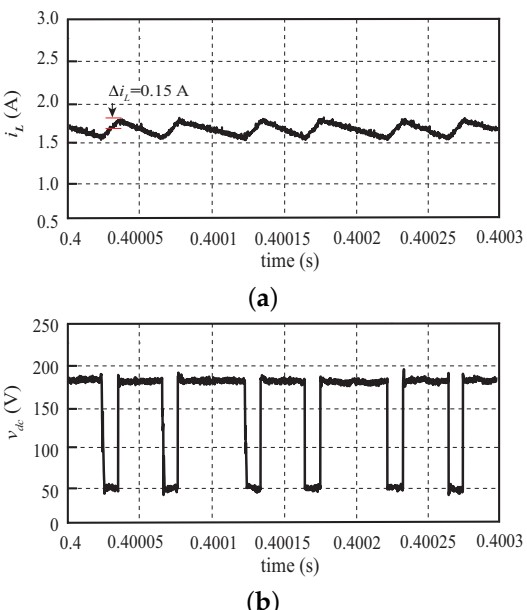

**Figure 11.** Z-source network waveform. (**a**) Inductive current (**b**) DC link voltage.

Figure 12a displays the output voltage waveform and output voltage error of the PCC point when PI control is employed. The output voltage waveform distortion of the PCC point is clearly visible, the steady-state inaccuracy is the biggest, and the tracking ability is subpar as can be seen from the figure. Figure 12b displays the PCC point voltage waveform and output voltage error when RC control is utilized. Compared to PI control, the voltage waveform is smoother, but the steady-state error is better. However, the dynamic response time is slower. Figure 12c displays the PCC point voltage waveform and output voltage error when utilizing DPRC control. Compared to the first two control strategies, the output voltage waveform and steady-state error are both superior.

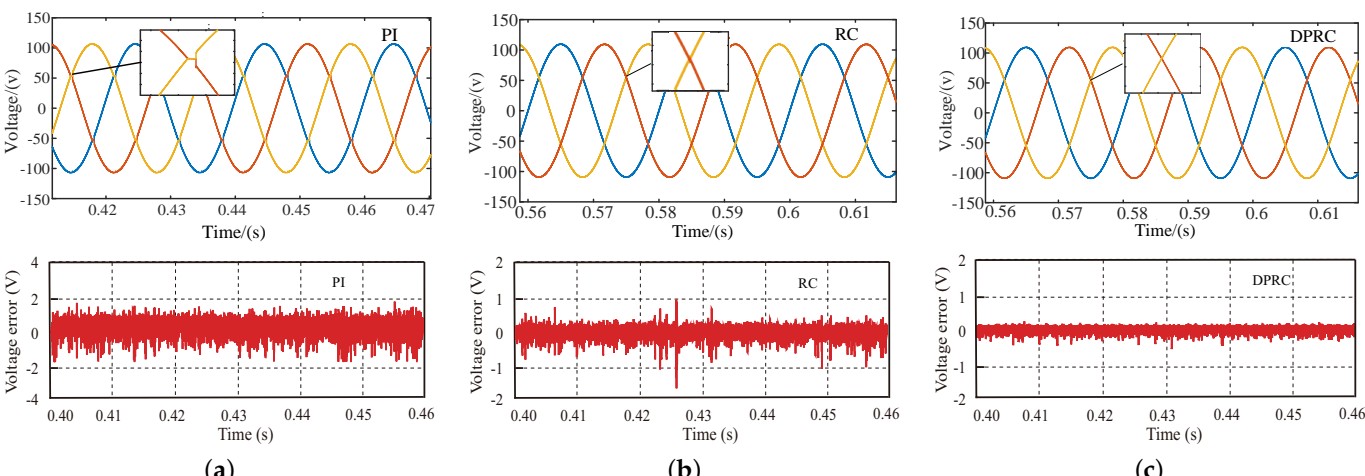

**Figure 12.** PCC output voltage and d-axis voltage error. (**a**) PI controller (**b**) RC controller (**c**) DPRC controller.

The overall simulation duration when DG1 and DG2 inverters are working in parallel is 1 s, the switch is detached in 0.4 s, and DG2 stops operating. Figure 13 waveform display, when the microgrid operates in parallel, the PCC voltage is virtually constant and maintains the sine curve, and the distribution and conversion of active and reactive power also retain high accuracy throughout the process. It can react fast and reach a new stable state in this manner.

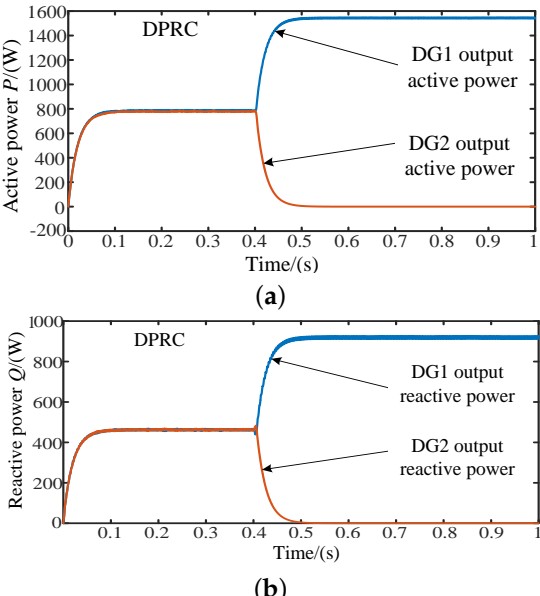

**Figure 13.** DG output active and reactive power. (**a**) DG output active power (**b**) DG output reactive power.

*4.2. Experimental Results*

To further confirm the accuracy and viability of the suggested control technique, the prototype of LC off grid inverter system was built using the experimental equipment of Bronze Sword Technology Company, as shown in Figure 14. The inverter system adopts three-phase full bridge circuit, and the hardware includes controller, drive protection circuit and sampling circuit. The experimental control chip adopts the TMS320F28335 chip produced by TI Company, and the PWM wave is generated by the ePWM module of the TMS320F28335 chip. The parameters of the main circuit and the controller are consistent with the simulation.

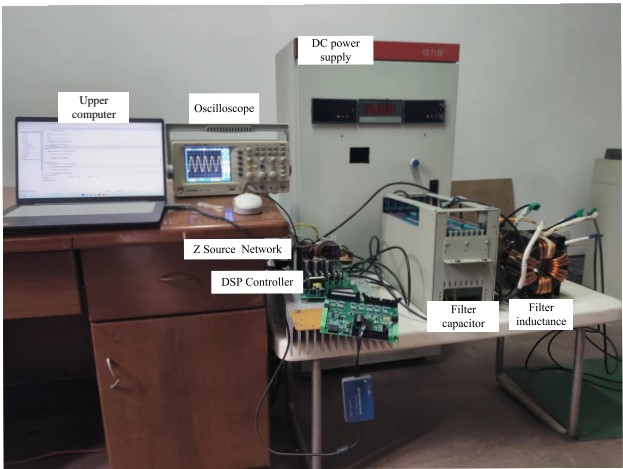

**Figure 14.** Z-source inverter experimental platform.

This paper sets a load switch from no load to full load to verify the dynamic performance and stability of the system under the most extreme conditions. At the same time, in order to verify the performance of the proposed deadbeat predictive repetitive controller, the PI controller and the repetitive controller in dq coordinate system are compared under the same working condition and load condition, and the validity of the control strategy proposed in this paper is verified.

Figures 15 and 16 shows the voltage and current waveform under linear and nonlinear load using PI control, repetitive control and deadbeat predictive repetitive control in dq coordinate system, and Fast Fourier Transform analysis of steady-state voltage. In the diagram, $i_a$ and $v_{aN}$ are the output current and voltage of phase a inverter respectively.

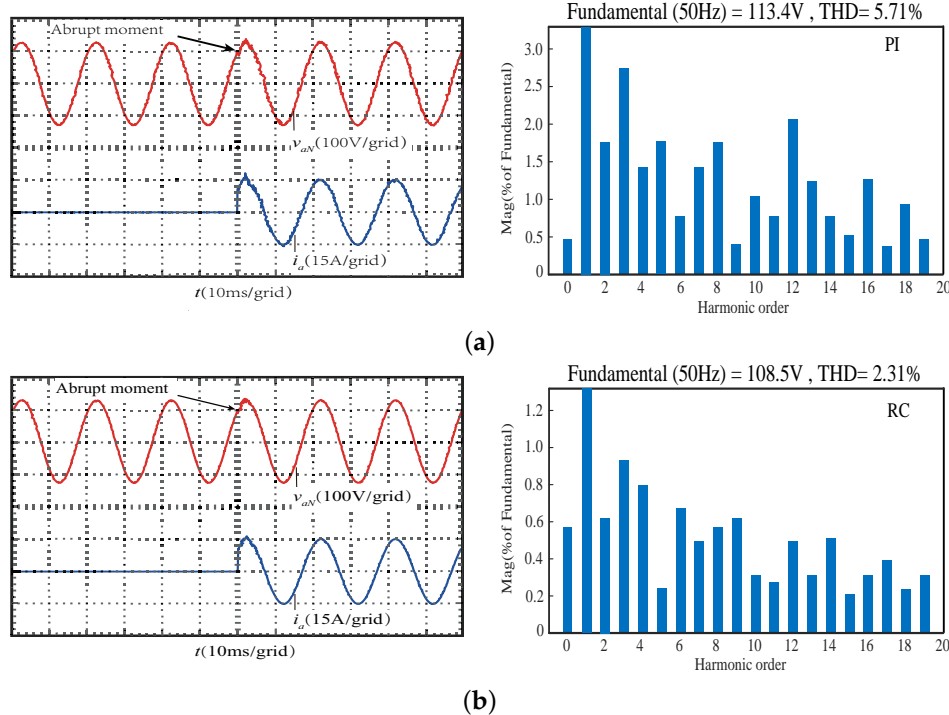

**Figure 15.** *Cont.*

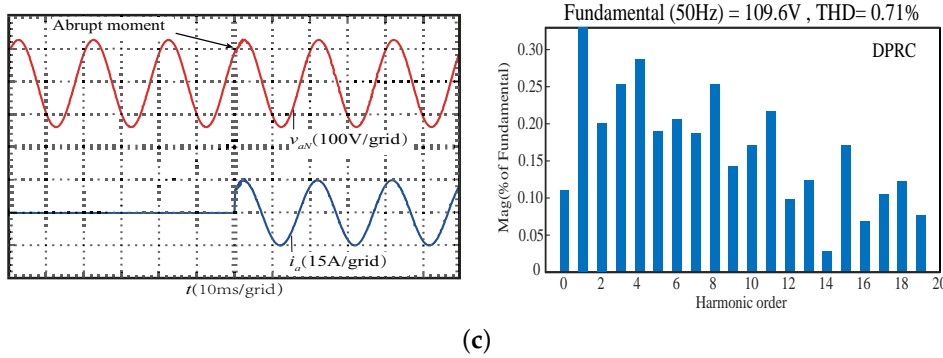

**(c)**

**Figure 15.** Voltage, current and voltage waveform distortion under linear load. (**a**) PI controller (**b**) RC controller (**c**) DPRC controller.

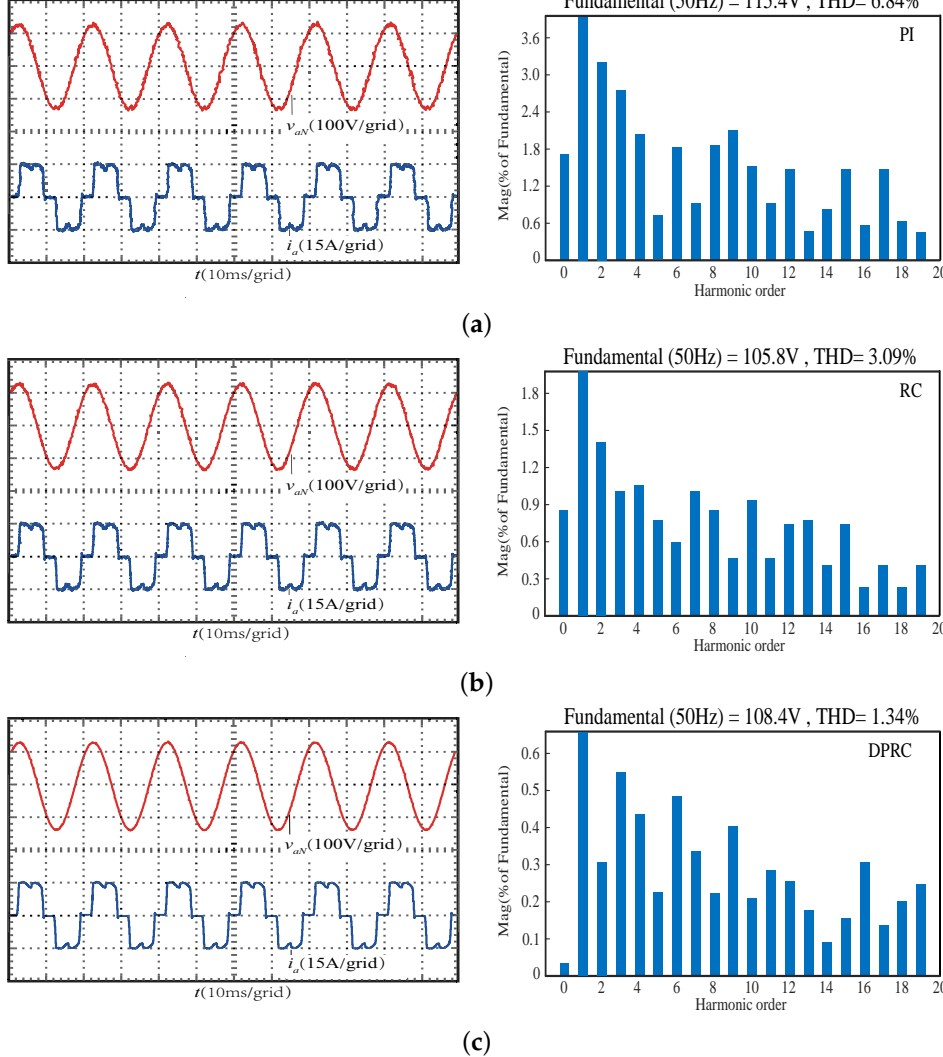

**Figure 16.** Voltage, current and voltage waveform distortion under nonlinear load. (**a**) PI controller (**b**) RC controller (**c**) DPRC controller.

Under linear load, when the inverter is controlled by PI, it can be seen from Figure 15a that the adjustment time is 10 ms and the total harmonic distortion rate of the output voltage is 5.71%. Figure 15b,c show that the dynamic response of the repetitive controller is basically the same as that of the control strategy proposed in this paper. The output voltage reaches steady state after 3 ms of linear load input, and the total harmonic distortion rate

of the output voltage is 2.31% and 0.71%, respectively. Under non-linear load, when the inverter is controlled by PI, the output voltage quality is poor under non-linear load, and the total harmonic distortion rate of output voltage is 6.84%. Figure 16b,c show that the dynamic response time of a nonlinear load is about one wave period due to the presence of a supporting capacitance, which takes longer to adjust than a linear load. The THD of the output voltage of the inverter does not exceed 5% at steady state. By introducing the state feedback of capacitive current and output voltage, the proposed control strategy further improves the system stability compared with the capacitive current feedback of repetitive controller. The total harmonic distortion of the output voltage of the inverter with repetitive control and beat-free predictive repetitive control is 3.09% and 1.34%, respectively. Compared with Figures 15 and 16, the control effect of the proposed control strategy is better than that of the repetitive controller under both load conditions, which shows that the proposed control strategy has good dynamic and steady-state performance for both linear and non-linear loads.

## 5. Conclusions

When the voltage distortion at the common grid-connected point of an isolated microgrid is caused by the non-linear load of the microgrid inverters, the traditional PI control method is insufficient. A composite control method is presented in this paper. Combining the advantages of Z-source inverters in boosting function and improving the system energy conversion rate through the through state with the linear matrix inequality design method, the design conditions of state feedback control law based on linear matrix inequality are obtained by using Lyapunov theory, the controller parameters are solved by linear matrix inequality, and the repetitive control and deadbeat predictive control are combined to design the corresponding simplified deadbeat predictive controller. The validity of the proposed control method is verified by simulation and experiment. The experimental results show that the proposed control strategy has good robustness under both linear and nonlinear loads.

**Author Contributions:** Project administration, F.P.; writing—original draft preparation, W.X. and J.Y. The final manuscript has been approved for publication by all authors after reading it. All authors have read and agreed to the published version of the manuscript.

**Funding:** This research was funded by the Natural Science Foundation of Hunan Province under Grant 2020JJ6019 and 2021JJ50115; Key project of Hunan Provincial Department of Education 20A116.

**Data Availability Statement:** Data are contained within the article.

**Conflicts of Interest:** The authors declare no conflict of interest.

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
