# Peer review of "State Feedback and Deadbeat Predictive Repetitive Control of Three-Phase Z-Source Inverter"

_electronics, doi:10.3390/electronics12041005_

Round 1

Reviewer 1 Report

The subject of the paper is interesting. The manuscript is well planned however, the Authors should pay attention to some aspects listed below:

- In the introduction section, Authors should clearly highlight the novelty of their work

- The analysis should be supplemented with a comparison of the obtained results with literature references and results of other researchers

- selected fonts in Figure 6 should be larger, because they are illegible

- references to Figures in the text should be written uniformly, for example, in line 208 is "Fig.11", and differently in line 218 "Figure 12"

- Description signs in Figure 11(a) and Figure 15 are illegible and should be corrected

- Figure 16 is placed between literature references Authors should correct such editing errors

Author Response

1- In the introduction section, Authors should clearly highlight the novelty of their work

Author response: Thank you for your comments on the draft. We have revised the abstract to enhance the description of the novelty of the article. The first sentence of the abstract reflects the novelty of the article.

2- The analysis should be supplemented with a comparison of the obtained results with literature references and results of other researchers

Author response: Thank you for your comments on the draft, which is not compared with specific references, but the methods used in this paper are compared with those in the references, and the experimental section is also compared with other methods to further verify the effectiveness of the methods proposed in this paper.

3- selected fonts in Figure 6 should be larger, because they are illegible

Author response: Thank you for your comments on Figure 6. Admittedly, some parts of the font in Figure 6 need to be enlarged to see clearly. Therefore, we choose a larger font for Figure 6. For specific changes, see the revised manuscript

4- references to Figures in the text should be written uniformly, for example, in line 208 is "Fig.11", and differently in line 218 "Figure 12"

Author response: Thank you very much for pointing out that there are inconsistencies in the expression in this article. We have revised them while checking the relevant expressions in the full text and revising them.

5- Description signs in Figure 11(a) and Figure 15 are illegible and should be corrected

Author response: Thank you for your comments on the graph. We have corrected the descriptive symbols in Figure 11 (a) and Figure 15 to make the whole more readable.

6- Figure 16 is placed between literature references Authors should correct such editing errors

Author response: Thank you for your comments on the typesetting of the manuscript. Figure 16 is between the references, which is really inappropriate. Therefore, we have modified the typesetting. For details, see the revised manuscript.

Reviewer 2 Report

To lower the harmonic content of output voltage utilising a Z-source inverter, a combined control method combining repetition control (RC) and deadbeat 1 predictive control (DPC) is proposed in this paper. There are some comments have to be done to enhance the paper body and results. These comments are:

1- Eq. (4) UC... C has to be in Capital.

2- There is something incorrect with all of the voltage spectra figures (Figs. 15 and 16). The fundamental appears in these figures with order 2 rather than 1.

3- Explain in detail and depth the repetitive controller.

4- What are the values of the gains (Ks) in state-feedback control?

5- Static loads (linear and nonlinear) have been tested; what about dynamic loads? (e.g: Induction motor).

6- Update the references list (2020 and above).

Author Response

The weight of the paper has been reduced, and the revised manuscript has been uploaded.

Round 2

Reviewer 2 Report

Thanks for responding to some of my comments. Still there are two comments author(s) have to reply to them.

1- The spectra analysis in Figure 15 and 16, author(s) zoom in so it shows all the even harmonics rather than odd harmonics. IF you please show us the complete spectra without any zooming considering 50 Hz has an order of 1.

2- All tests were performed with static loads (linear and nonlinear), but no dynamic loads were tested to evaluate the controller. A simulation figure will be added to test the system under dynamic conditions.

Author Response

1- The spectra analysis in Figure 15 and 16, author(s) zoom in so it shows all the even harmonics rather than odd harmonics. IF you please show us the complete spectra without any zooming considering 50 Hz has an order of 1.

Author response: Thank you for your comments on the pictures of the manuscript. After careful check, we have made corresponding amendments. For details, please see the revised manuscript.

2- All tests were performed with static loads (linear and nonlinear), but no dynamic loads were tested to evaluate the controller. A simulation figure will be added to test the system under dynamic conditions.

Author response: Thank you for your question, which makes sense. Firstly, since this paper mainly focuses on the novel control algorithm for the power quality problem of off-grid inverter, only considering the linear non-linear load, or considering the increase of unbalanced load, while the dynamic load like induction motor belongs to a new research content, involving the control of motor drive system. If this part is added, it will appear that the article is more complex and may not be able to return the revised version by the deadline. For this article, the simulation of Figure 13 is about the effect of switching action of DG1 and DG2 on output power. This part can verify that good performance also exists in system action switching.
